# Research on Single Point Incremental Forming Characteristics of Perforated TA1 Sheet

**Ruxiong Li** [1,2,*] and **Tao Wang** [1]

1 College of Material Science and Technology, Nanjing University of Aeronautics and Astronautics, Nanjing 211100, China
2 School of Mechanical Electronics Engineering, Jingdezhen Ceramic University, Jingdezhen 333403, China
* Correspondence: langxue3000@163.com

**Abstract:** In view of its high compactibility, flexibility, reduced compacting pressure, and other superiorities, the single point incremental forming (SPIF) technique has been applied increasingly in the cranial prosthesis forming of perforated TA1 sheet with irregular surface features. Whereas the SPIF of perforated sheet, which seems to meet many challenges in the biomedicine field, where control of component geometric dimensions are qualification requirements of cranial prosthesis. The mechanism of incremental forming and characteristics of perforated TA1 sheet become more complicated because of the mesh apertures. Hence, certain path and forming technique parameters were set to conduct finite element modelling on the truncated right angle cone components of perforated TA1 sheet and titanium plate respectively on the basis of the ANSYS/LS-DYNA platforms, in light of the constructed model, the distribution and variation law of displacement field coupled with the contour accuracy, wall thickness, and strain capacity for different areas of components in the forming process were obtained by researches, bringing up to light the forming mechanism of conical perforated TA1 sheet components. The research findings indicate that the aperture structure of perforated TA1 sheet allows the material elements to accomplish deformation in the surface where the aperture is located, and the strain rate and radial aperture growth rate of perforated titanium sheet are relatively high, the maximum radial aperture growth rate value is as high as 78.53 percent, the maximum circumferential aperture growth rate value is only 10.84 percent, the extension-thinning of forming section for perforated TA1 sheet is higher than that of the titanium plate, and perforated TA1 sheet possesses higher geometric accuracy than titanium plate.

**Keywords:** single-point incremental forming; perforated TA1 sheet; forming strain; thinning ratio; numerical simulation

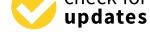



## 1. Introduction

Due to its good biocompatibility, the titanium alloy is regarded as the acceptable materials for biomedical application (implantation materials and prosthesis equipment). The customization of repairosome has become a focused field in respect to biomedical engineering. Comprehensive analysis has been conducted on the main types as well as the major merits and demerits of titanium alloy adopted in medical application by Rack H.J [1] and some other people. Geetha M [2] has elaborated on the best solution for the problem of orthopedic implantation regarding titanium alloy. Aydin et al. [3] have reviewed the techniques and materials applied in cranioplasty at great length. Numerous scholars have pointed out [4] that titanium alloy is one of the preferred materials of cranial prosthesis for large cranial defects, but it also has difficulty forming.

Investment casting used to be an alternative method for manufacturing restorations, complex shape of perforated titanium sheet before casting, need to optimize the porous shape of perforated sheet, after casting need to treat the surface of the formed parts, design and processing costs are relatively high [5], besides, titanium reacts very strongly at high temperatures, it is difficult to achieve the formation of long thin-walled restorations.

Due to the relatively limited production volume of titanium and its alloys, additive manufacturing technology has been used to shape different anatomical structures in recent years, such as the hips, skull, cheekbones, mandibles, and other medical restorations. However, the materials required by additive manufacturing technology rely on imports, and the problems of insufficient material performance and high cost still need to be overcome, in addition, the lack of evidence-based technical standards for the biomechanical properties of 3D printing products, the clinical system for printed products is not perfect. Therefore, how to cost-effectively manufacture high-quality titanium skull restorations is a new challenge in the field of metal forming.

Incremental sheet metal forming in general and SPIF specifically have gone through a period of intensive development with growing attention from research institutes worldwide [6–8]. The main design factors for incremental plate forming are plate thickness, material properties, and part shape, with low cost and cycle times, making them ideal for the molding of custom parts with complex geometry in small batches [9–12]. After early exploration of the potential of single point incremental formation (SPIF) in the manufacture of skull restorations [13–15], Fiorentino et al. [16] raised questions about the biocompatibility of the repairosome and surface finishing and treatment in their preliminary studies. Ambrogio et al. [17] used the incremental forming process to produce a customized ankle stent. By scanning the ankle joint of patient and generating a virtual model from the scanner output information (point cloud), a virtual model was used to create an ankle joint stent on the precise size of patient through an incremental shaping process. Göttmann et al. [18] discussed a preliminary study of titanium cranial prosthesis manufacturing using a two-point incremental forming process. Olesik [19], Eksteen et al. [20] have also proposed the use of incremental prototyping to produce non-epicondylar knee arthroplasty restorations. Racz and some other people [21] used the decision-making method of AHP (analytic hierarchy process) to compare several methods of processing TA1 titanium skull restorations (cranioplasty plates), and by comparing formability, microstructure, control, roughness, energy consumption, accuracy, and production time, the results showed that SPIF at room temperature is the best way to manufacture medical devices. However, titanium alloys that undergo plastic deformation at room temperature are low-plastic materials, so special measures must be taken. Therefore, incremental molding has become a custom biomedical parts manufacturing process with important application prospects.

The above research is mainly for the incremental formation of titanium plate skull restoration, and for the purpose of facilitating the growth of granulation tissue between the skin and dura mater, the interconnected multi-directional micro apertures within the interior produce capillary permeation, and most of the skull restorations currently use perforated titanium sheet. The SPIF of perforated titanium components are achieved by the continuous variation of local deformation accumulation between the layers. However, there is a difference in physical and mechanical properties between titanium plate and perforated titanium sheet, and the mesh aperture domain will flow in a circular direction during the incremental forming of perforated titanium sheet, which is more complex than the degree of deformation of titanium plate. Due to the limitations of experimental means and analytical tools, the research on the incremental forming mechanism and process of perforated titanium sheet is less and not systematic. Most SPIF work applied in biomedicine field has not been developed on perforated sheets. It is in this context that this research was conducted to produce perforated cranial prosthesis.

Experiment research is the basic method to explore the SPIF law of perforated titanium sheet. Bouzidi [22] has shown the feasibility of incremental forming tests on perforated sheets of AZ31B Magnesium alloy. Baik [23] and Elangovan [24] have shown that the formability of perforated sheet depends on the ligament rate of perforations and loading state. For the features of small strain and large deformation in the SPIF process, there are many deformation modes in the SPIF process, such as shear, drawn, and bending. In addition, due to the thin wall thickness of the perforated sheet, the edge of the perforations are easy to overthin in the forming process, and the final deformation and instability rupture.

All of these lead to the bad forming performance of the perforated sheet. Therefore, it is very important to study the SPIF law of the perforated sheet and analyze the possible forming defects. However, the SPIF mechanism of perforated sheet cannot be clearly explained by experimental studies and simplified theoretical analysis.

Therefore, by means of finite element simulation, aiming at the cone piece models of perforated TA1 sheet, compared with the incremental forming of the titanium plate, the incremental forming mechanism and deformation characteristics of perforated titanium sheet with the perforated titanium sheet of the equilateral pattern arrangement were studied in this paper, particular attention was paid to the distribution and variation law of displacement field coupled with the contour accuracy, wall thickness, and strain capacity for different areas of components, which provided theoretical guidance for the incremental shaping of the perforated TA1 sheet skull restoration.

## 2. Construction of Finite Element Model

### 2.1. Construction of Material Constitutive Model

Titanium plates generally have anisotropic characteristics, therefore, for the purpose of establishing a reasonable TA1 titanium alloy (which are certificated for medical applications according to the DIN ISO 5832-2 norm) constitutive model of 1 mm, by water jet cutting at an angle of 0°, 45°, and 90° from the rolling direction of the plate to obtain three groups of tensile samples in different directions [25]. According to that set out in Standard GB/T 228-2010, normal anisotropy coefficients were obtained for each rolling direction by measuring the corresponding dimensions with a profile projector, as presented in Figure 1. The strain hardening index, strain hardening coefficient, elastic modulus, and thickness anisotropy coefficient of titanium alloy materials are obtained by tensile tests, which will be used to define the elastoplastic behavior of materials in finite element simulation.

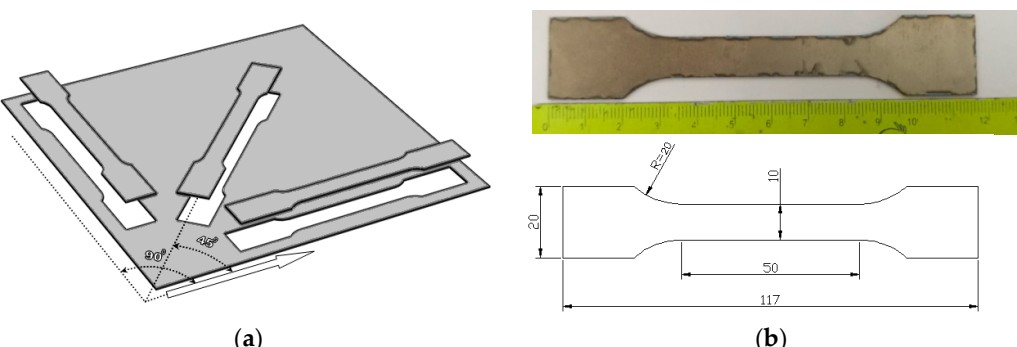

| (a) | (b) |

**Figure 1.** TA1 titanium alloy plate tensile test sample. (**a**) Locations of samples in different directions. (**b**) Tensile sample size (in mm).

Applying the SAN universal material testing machine (SSANS Inc., Shanghai, China), the loading tensile rate is 2 mm/min, and after data processing, the conventional strain curve of TA1 titanium plate at room temperature is obtained σ and elongation curve εmax, as presented in Figure 2. The mechanical properties of the materials determined by the tensile tests of the three samples are presented in Table 1, and the obtained material parameters are fed into the finite element simulation software (ANSYS/LS-DYNA, ANSYS and LSTC, Pittsburgh, PA, USA), to set up a constitutive model of titanium plates considering anisotropic properties.

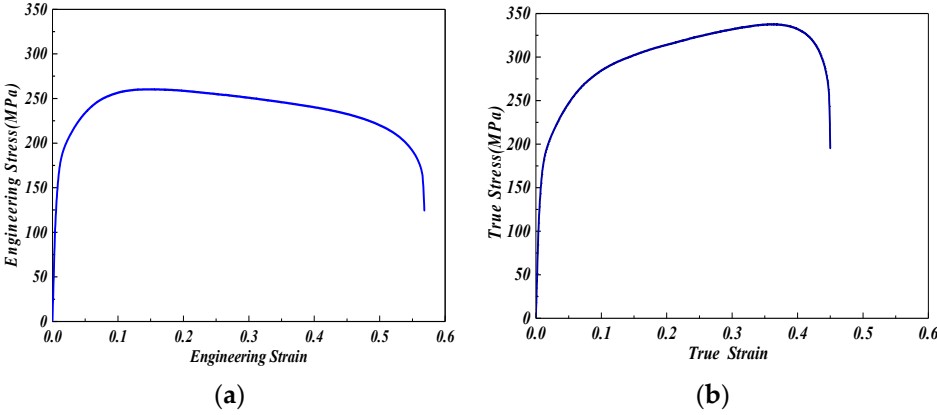

**Figure 2.** Stress strain diagram of the sample stretch. (**a**) Engineering stress–strain diagram. (**b**) True stress–strain diagram.

**Table 1.** Synthesis of data from tensile tests.

| Sample Cutting Angle | 0° | 45° | 90° |
|---|---|---|---|
| Modulus of elasticity E/GPa | 109.59 | 109.32 | 110.18 |
| Hardening coefficient n | 0.1175 | 0.1325 | 0.138 |
| Resistance coefficient K/Pa | 520.72 | 525.66 | 513.95 |
| Thick anisotropy index r | 0.796 | 2.661 | 0.669 |

### 2.2. Finite Element Modal and the Setting of Forming Parameters

For the purpose of ensuring the deformation stability of perforated titanium sheet during incremental forming, the mechanism of incremental forming of a perforated titanium sheet is systematically studied, and a finite element model with the goal of truncating tapered components is established in this paper, as presented in Figure 3. The process parameters used in this study are presented in Table 2, using a contour toolpath forming strategy with a hemispherical tool head end, the tool metal is W6Mo5Cr4V2 high–speed steel. The tool, the blank holder, and the backing plate were treated as rigid bodies. Shell elements (shell163) were selected to mesh the perforated sheet and solid elements (solid164) were selected to mesh the tool, the 1.5 mm size free grid were used for grid division. In order to more accurately describe the bending effect of the plate deformation and the stress and strain changes in the sheet thickness direction, as far as the utilized shell element was regarded, a 4–node plate, characterized by 5 integral points through the thickness and hourglass control, was chosen. The Belystchko–Wong algorithm was adopted. Coulomb's friction law was applied with a friction coefficient of 0.1 between the perforated titanium sheet and the tool. Additionally, the contact condition was implemented through a pure master-slave contact algorithm. During the SPIF process of perforated sheet, nodal displacement and rotation were constrained on all edges of the sheet.

**Table 2.** Perforated titanium sheet incremental forming process parameters.

| Parameter Name | Initial Wall Thickness t0/mm | Tool Diameter dp/mm | Wall Angle θ/° | Feed Velocity v2/(mm/min) | Step Down s/mm | Forming Depth h/mm | Aperture Diameter/mm | Hole Spacing /mm |
|---|---|---|---|---|---|---|---|---|
| Numeric value | 1 | 10 | 45 | 1000 | 1 | 25 | 4 | 8 |

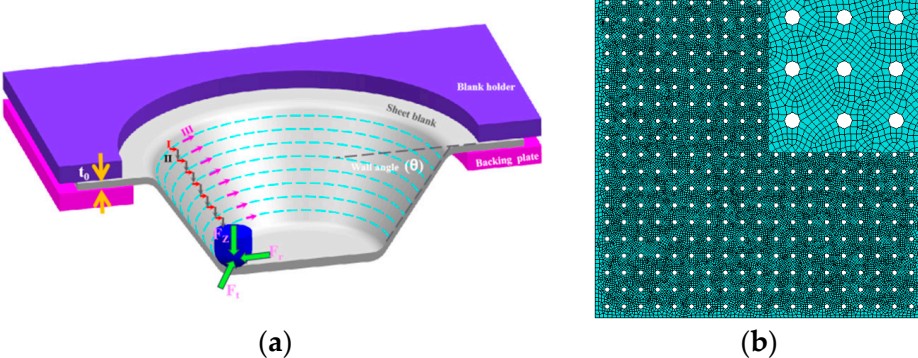

**Figure 3.** Schematic diagram of the incremental forming process of perforated titanium sheet. (**a**) Finite element modal. (**b**) Grid partition of the perforated titanium sheet.

The influence of bending deformation and material strain hardening of perforated titanium sheet is considered in the model, and the deformation stability of the incremental forming process under planar strain conditions is analyzed.

## 3. Results and Discussions

Figure 4 is a comparison diagram of the actual assembly wall thickness and the simulated shaped wall thickness of the fixed–angle cone–shaped piece. A total of three sets of process experiments and simulation experiments were done. The wall angle is respectively set to 10°, 25°, 45°. From Figure 4, it can be clearly observed that the error of the experiment and simulated values is 3.2%, when the wall angle is 10°, the error is 1.5% and when the wall angle is 25°, the error is 1.1%. When the wall angle is 25°, neither of the values exceed 5%, so it is believed that the experimental results are in line with the simulation results, clearly showing the reliability of the simulation.

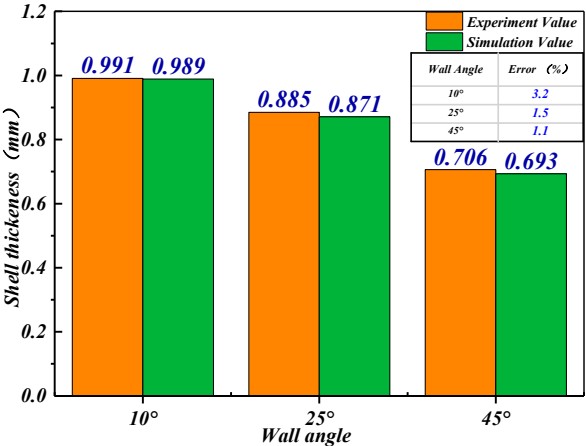

**Figure 4.** Comparison of the experiment and the simulation value of wall thickness error.

### 3.1. Strain Distribution Law of Perforated Titanium Sheet

For the purpose of analyzing the forming process of perforated titanium sheet and titanium plate, having a knowledge of the evolution of strain components, the contour forming path with a z–direction feed of 0.5 mm is selected, and LS–DYNA is applied to simulate SPIF of a truncated cone piece with an opening diameter of 80 mm. As presented in Figure 5, the deformation history of elements A, B, C, D, E, F, G and H along the outline of the deformation shape of the perforated titanium sheet in the simulation begins. Perpendicular to the tool movement and the direction along the oblique wall is defined as one, the forming direction (tangential direction) is defined as two, and the thickness direction is defined as three.

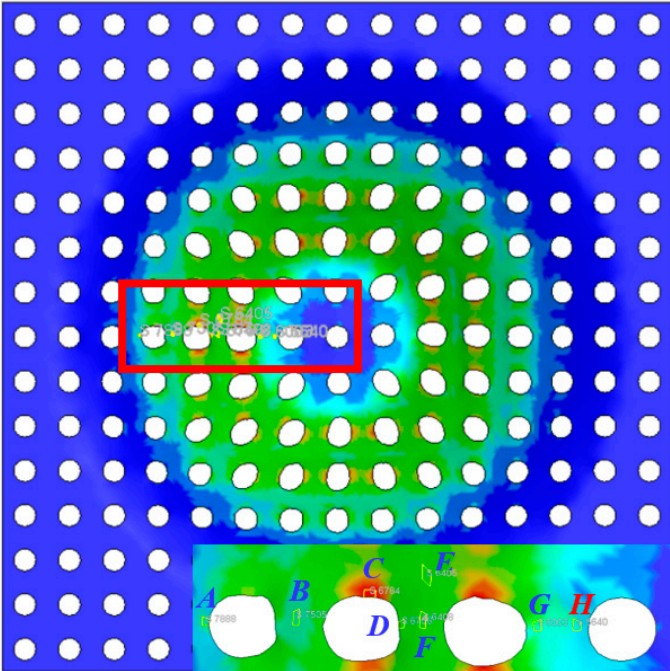

**Figure 5.** Schematic of sections A, B, C, D, E, F, G and H along the component wall at which the deformation history from FEA is examined.

Deformation in the initial stages of the perforated titanium sheet incremental forming process is due to bending, while further deformation is due to shear and/or stretching. The observed thickness values in the wall region (area BC) are very consistent with the sine law. In the main deformation area, the deformation of the incremental forming process of the perforated titanium sheet mainly comprises the extrusion $\varepsilon11$ perpendicular to the toolpath and the thickness shear perpendicular to the direction of the toolpath and along the cutter path, the main mechanism of the shear of the incremental forming process is in the accumulation of the material in front of the tool head, therefore, the tool drags the material along the trajectory direction to cause the penetration thickness shear $\varepsilon13$, and the absolute value of the shear strain $\varepsilon13$ parallel to the tool trajectory is greater than the shear strain $\varepsilon23$ perpendicular to the tool trajectory. The changes in perforated titanium sheet can be divided into three special areas:

(1) The perforated titanium sheet is red on both sides of the circular aperture short axis (tool movement circumferential direction) area, as presented in the element C of Figure 6 $\varepsilon11$ and $\varepsilon33$ gradually decay after rapid increase, indicating that under the action of the tool head, it tends to stabilize after stretching along the wall direction, $\varepsilon22 \approx 0$, $\varepsilon33$ first increases and then decreases, indicating that under the action of the tool head, the trend is stable after compression along the thick direction, and the shear strain $\varepsilon13$ tends to stabilize after rapid increase, indicating that the elements on both sides of the short shaft are plasticly sheared (through thickness shear).

(2) In the area of the long axis side (wall direction) of the circular aperture, as presented in Figure 5, of element B, element D, element F, and element G, the metal flows along the long axis direction and gradually accumulates, and the thickening phenomenon occurs. Moreover, $\varepsilon11$ is slightly increased to a negative value after a positive value, indicating that it is compressed along the wall direction, and $\varepsilon22$ tends to stabilize after rapid increase, indicating that it is stretched along the forming direction. Additionally, $\varepsilon33$ first increases and then decreases, indicating that it is compressed along the thick direction, and the shear strain $\varepsilon13$ tends to stabilize after rapid increase. Indicates that the elements on both sides of the short shaft are plasticly sheared (by thickness) in thickness.

(3)    At the bottom of the circular truncated cone, the cloud map can also shows that the thickness of the material plate is slightly increased, and the thickness of the sheet metal on the right wall is slightly larger than that of the side wall of other main deformation areas. This indicates that the sheet material in this area produce a one-way sheet accumulation phenomenon in the forming process, as presented in element H of Figure 6, where $\varepsilon 11$ is slightly increased. This indicates that it is compressed along the wall direction. The values $\varepsilon 22$ and $\varepsilon 33$ are larger, and tend to stabilize after rapid increase, indicating that it is stretched along the forming direction, compressed along the thick direction, and the shear strain $\varepsilon 13$ and $\varepsilon 23$ are slightly increased. It is plasticly sheared (through the thickness) in a thickness in the opposite direction to the main deformation zone.

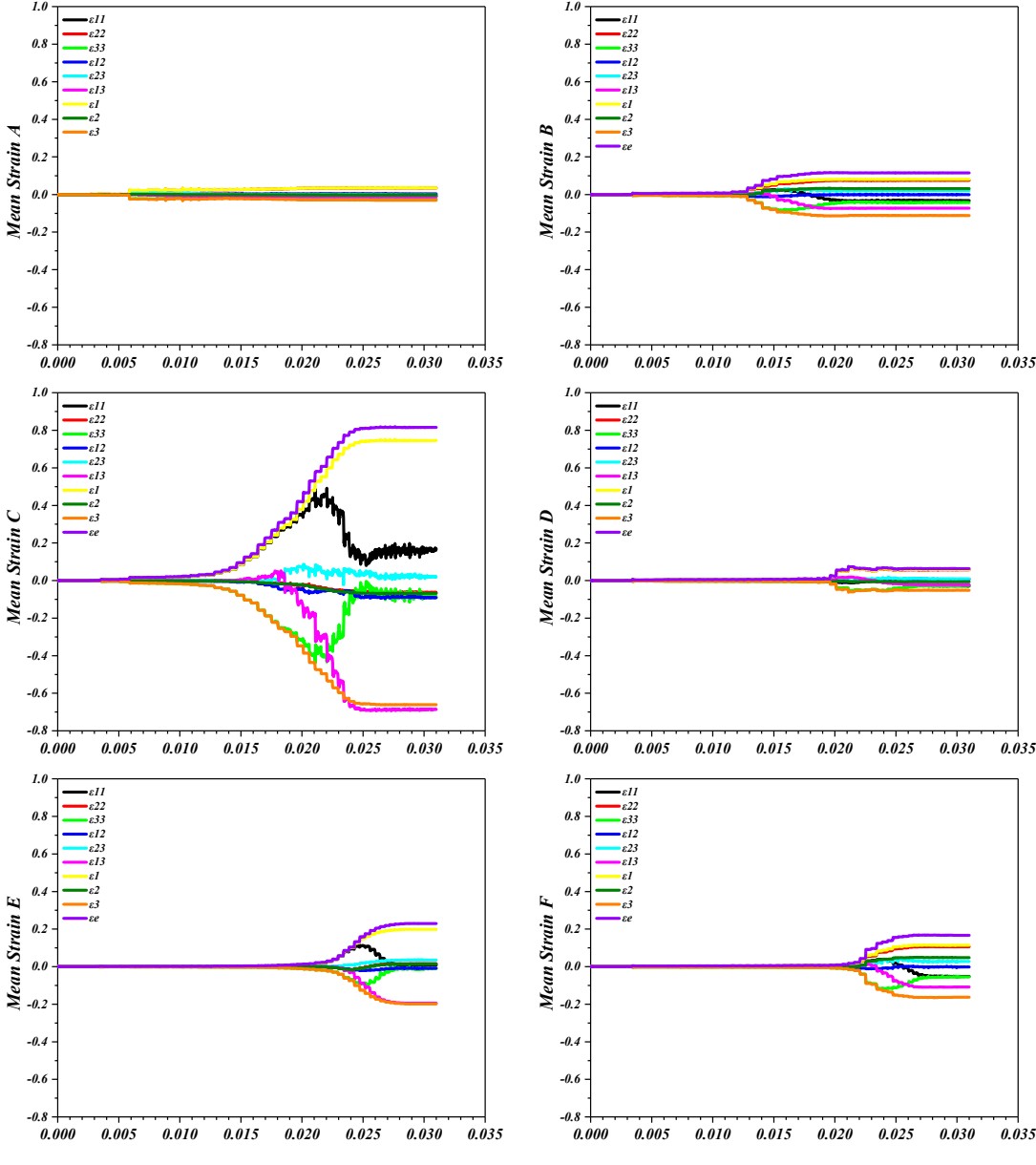

**Figure 6.** *Cont.*

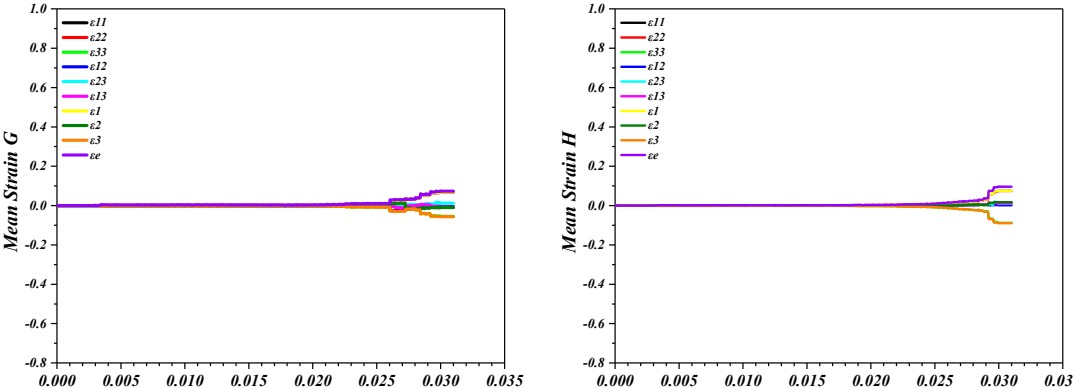

**Figure 6.** Evolution of strain at sections A, B, C, D, E, F, G and H.

Except for the above three special areas, element A is in the bending deformation zone, compared with other strains, $\varepsilon22$ and $\varepsilon33$ values are larger, and tend to stabilize after rapid increase, indicating that it is stretched along the forming direction and compressed along the thick direction. As presented in Figure 6 of element H in an area far from the mesh aperture, the deformation is similar to that of the short shaft side, but the value is smaller than the short shaft side. According to the three–way stress state, in the bending deformation area, the two sides of the short axis of the circular aperture in the main deformation area and the main deformation area away from the mesh side element, $\varepsilon2 \approx 0$, $\varepsilon1 \approx \varepsilon3$, can be considered to be subject to shear action, and subject to plane stress state. At the bottom of the truncated cone piece and the main deformation area, $\varepsilon2 > 0$, $\varepsilon1 + \varepsilon1 \approx -\varepsilon3$, are in a typical two-way tensile and one-to-one compression mechanical state.

Figure 7 presents a histogram of the distribution of the maximum effective plastic strain values of elements A, B, C, D, E, F, G, and H of a perforated titanium sheet cone at the end of forming. In addition to the bending deformation area of the perforated titanium sheet, the effective plastic strain of the lower surface of the lower surface of the circular aperture area and the bottom upper surface of the truncated cone piece is larger than that of the lower surface, the effective plastic strain of the lower surface of the other areas is greater than the upper surface, which is due to the local stretching and bending around the tool, which makes the element stretch more than the upper surface, which causes the plasticity of the lower surface of the perforated titanium sheet to increase.

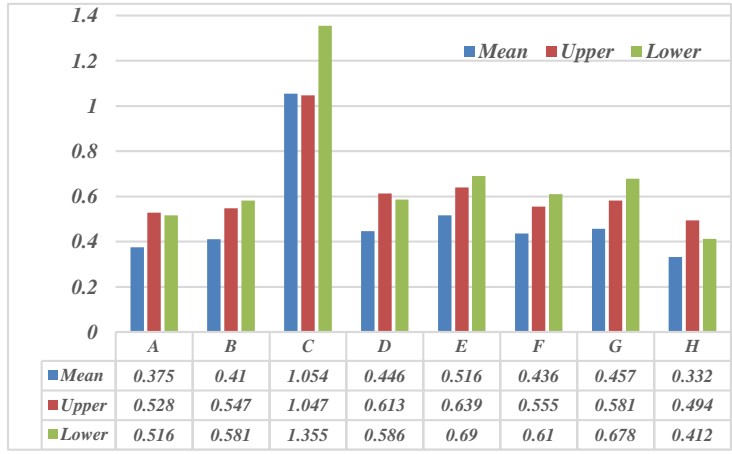

| | A | B | C | D | E | F | G | H |
|---|---|---|---|---|---|---|---|---|
| ■ Mean | 0.375 | 0.41 | 1.054 | 0.446 | 0.516 | 0.436 | 0.457 | 0.332 |
| ■ Upper | 0.528 | 0.547 | 1.047 | 0.613 | 0.639 | 0.555 | 0.581 | 0.494 |
| ■ Lower | 0.516 | 0.581 | 1.355 | 0.586 | 0.69 | 0.61 | 0.678 | 0.412 |

**Figure 7.** Histogram of the distribution of the maximum effective plastic strain values of the perforated titanium sheet at the end of forming.

### 3.2. Geometrical Comparisons

The major defect of incremental forming is the lack of geometric accuracy of the resulting product. Forming accuracy is an important indicator to measure the machined

part, and the geometric error of perforated titanium parts refer to the dimensional deviation of the target part from the design CAD model in the geometric space in vertical Z planes [25], as presented in Figure 8.

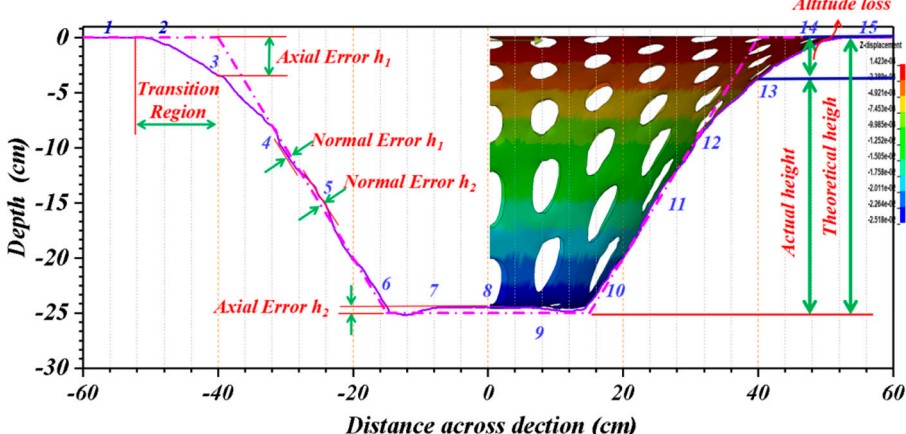

**Figure 8.** Geometric error evaluation diagram of perforated titanium sheet cone counterparts in vertical Z plane.

On the basis of the given shape, the feature edge of the CAD model and the starting forming surface should be at a set angle (45° in the figure), but there is a very discrepancy between the actual geometry and the desired one, the feature wall of the bending transition zone of the main deformation area of the perforated titanium sheet is highly curved and convex, and the eigen edge is obviously visible with a large deformation. Such a defect strongly depends on the absence of any support that may reduce the sheet bending. This paper uses the axial error h1 to measure the pull-down and bending deformation of the perforated titanium sheet, that is, the cantilever beam model will be formed between the initial forming surface of the perforated titanium sheet and the fixture. When the tool applies the force of axial downward pressure on the perforated titanium sheet, it is effective to applying a concentrated load at the end of the beam, and the cantilever beam undergoes bending plastic deformation under the continuous action of the concentrated load so that the height loss in the direction of the height of the formed part occurs h1.

In the major deformation area of the perforated titanium sheet and the contact deformation area at the bottom of the truncated cone piece, the illusion of forming a deep multi–CAD theoretical forming depth (25 mm in the figure) is generated, and the reason can be interpreted as: when the tool feeds the perforated titanium sheet vertically in the axial direction, the close contact between the tool and the sheet makes the perforated titanium sheet shape similar to the tool head. In addition, the titanium net falls into the knife mark so that a concave area is generated. At the bottom of the truncated cone piece, we find that the actual feature depth is slightly less than the theoretical forming depth (25 mm in the figure). The cause of this can be described as: when the side wall of the tool is rolled down axially, due to the law of the minimum resistance, the perforated titanium metal flows along the direction of the minimum resistance, accumulating on the lower side of the aperture and the bottom of the truncated cone piece. This process continues until the last layer is forming. At this time, there is no longer the reverse bending moment of the next layer of processing on the perforated titanium sheet, and the excess metal is under the action of bending stress. A deformation of the upward bump is produced, defined as the axial error h2 ("head in pillow effect") of the perforated titanium sheet.

In the formed area of the major deformation area, there is a deviation of both concave and convex in the side wall of the perforated titanium sheet, where the convex is defined as the normal error h1, and the concave is the normal error h2. After the repeated loading of the perforated titanium sheet by the head and the accumulation of a rebound, a curvature is generated on the expected straight line edge, that is, (normal error h1 and h2), which is related to the deformation characteristics of the discontinuous region and the continuous

region of the perforated titanium sheet. In the discontinuous region of the perforated titanium sheet, the perforated sheet element will be deformed under the action of the tool, in the form of convexity, and in the continuity area, especially the continuity region element near the lower part of the mesh element. The accumulation of the material makes it in a compressed state, therefore, manifested in the form of an inverted concave.

The geometric errors described above are also the geometric defect of the perforated titanium sheet. Although it is impossible to exactly reproduce the desired geometry, the errors entity may reduce if the process parameters can be accurately selected in the later stage.

The cross-sectional distortion rates are prominent indicators of forming quality. For the purpose of understanding the deformation quantity of the mesh aperture, the elements on the Y = 0 section of the perforated titanium sheet at the end of forming is also analyzed, and the radial and circumferential aperture diameter growth rate change curve presented in Figure 9 is obtained. It can be observed that the strain rate of the perforated titanium sheet is higher than that of titanium plate, so the radial aperture growth rate of perforated titanium sheet is larger than that the radial growth rate of titanium plate. The maximum radial aperture growth rate is as high as 78.53 percent, and the maximum circumferential aperture growth rate is also as high as 10.84 percent. The results show that the aperture size of perforated titanium sheet is mainly radial growth, which is much larger than the circumferential aperture size growth rate.

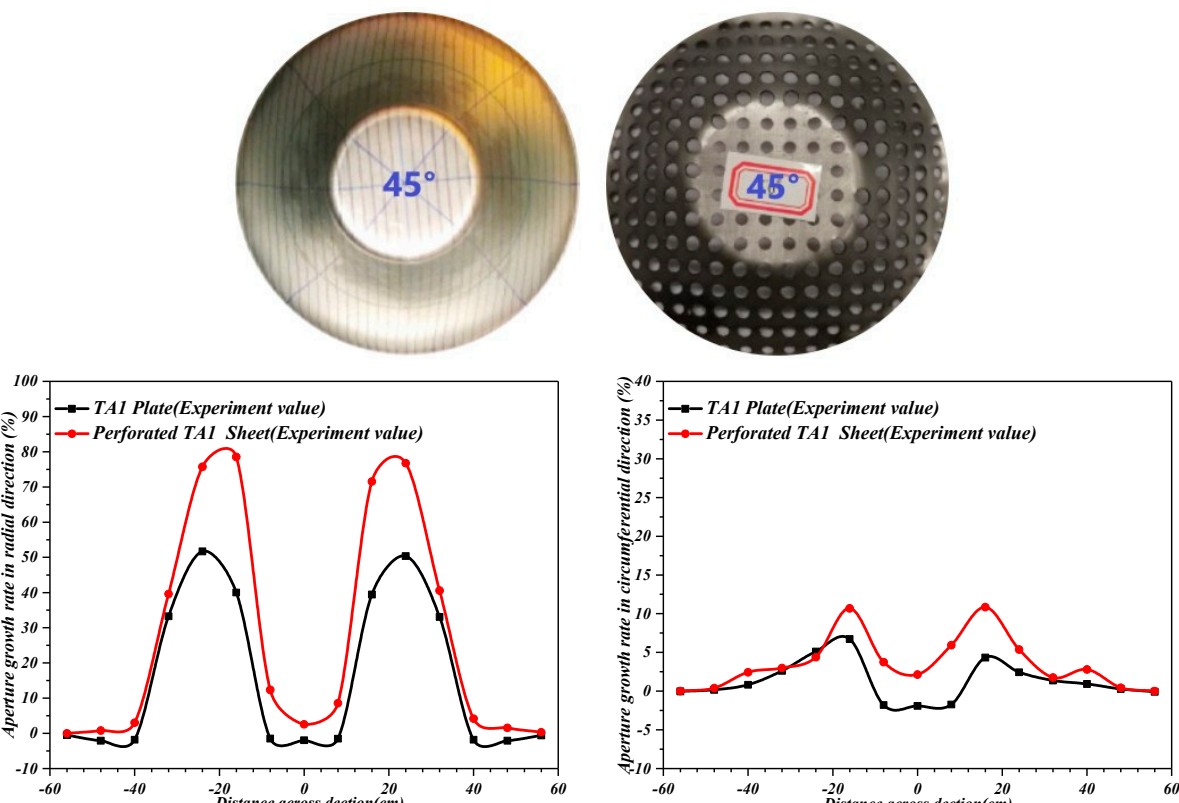

**Figure 9.** Curve of radial and circumferential aperture growth rate of titanium plate and perforated titanium sheet along the X axis.

### 3.3. Wall Thickness Evolution

During the incremental forming process, the thinning of the flakes is due to the accumulation of strain on contact with the tool. The thickness of the undeformed material blank is an important parameter that has a significant impact on the incremental forming process and the final part, especially the force required to deform the plate, which increases with the thickness. Assuming a constant volume and a deformation mode close to the

plane strain conditions, the actual thickness of the cone wall with an initial thickness of $t_0$ can be deduced from the sinusoidal law. Each combination of forming parameters causes the thickness to change, and the thickness value of the bending transition deformation zone is higher than the thickness corresponding to the deformed wall. The uneven thickness distribution depends mainly on the change in stiffness in the plate. In the bending transition deformation area, due to the high stiffness near the clamping area and the lack of back support, the deformation is low, and it is in a bent state from the beginning of deformation; in the center of the sheet, the lower stiffness value makes the region more deformed.

The mesh aperture of perforated titanium sheet follows the bottom of the mesh aperture in the direction of the forming wall, as well as the bottom area of the truncated cone piece, resulting in a slight increase in the final thickness due to the forming tool pushing the metal material with considerable plastic deformation down.

The decrease in wall thickness is due to the shear of $\varepsilon 13$ and the tensile action of $\varepsilon 11$, resulting in the metal being pushed to the center of the truncated cone piece. A detailed description of the evolution of the thickness of the perforated titanium sheet is crucial for accurate force prediction. As presented in Figures 10 and 11, the entire formation is achieved by thinning the deformation area, and the thickness distribution of the perforated titanium sheet in the XZ plane and the YZ plane is almost exactly the same, indicating that the part is formed correctly and maintains symmetry. The thinning strip of perforated titanium sheet is distributed in a layered square; The circular apertures in the main deformation area of the perforated titanium sheet have become elliptical apertures, and the thickness of the short axis side elements in the elliptical aperture are obviously longer. The thickness of the shaft side elements is small, and it can also be found from the strain energy map that the strain energy at the short axis can be negative, which can determine the compressive stress on both sides of the short axis. During the incremental forming process, the strain accumulation and intermittent stresses generated by the contact between the tool and the perforated titanium sheet affect the thinning rate of the perforated titanium sheet. Consistent with the strain of the perforated titanium sheet, the thickness change of the perforated titanium sheet can be divided into three special areas. In the first area, the perforated titanium sheet is blue on both sides of the short axis of the circular aperture (the circumferential direction of the tool movement), and the perforated titanium sheet is subjected to two–way pulling and unidirectional pressure. In the layer–by–layer forming process of the tool head, the perforated titanium sheet is exchanged for the thickness of the area, so that the thickness of the perforated titanium sheet is reduced and the surface area is enlarged. In the second, the area of the long axis side (wall direction) of the circular aperture, the metal flows along the long axis and gradually accumulates, and the thickening phenomenon occurs. In the third area, at the bottom of the truncated cone piece, the cloud map can also see that the thickness of the material plate is slightly increased, and the thickness of the sheet metal on the right wall is slightly larger than that of the side wall of other main deformation areas. This indicates that the sheet material in this area has produced a one–way plate accumulation phenomenon in the forming process.

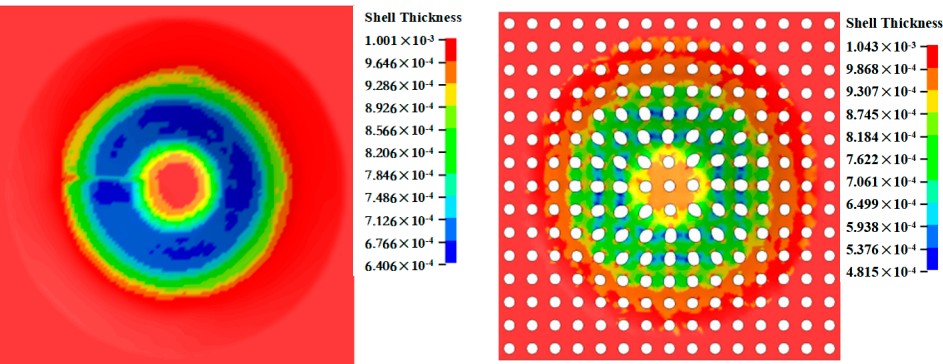

**Figure 10.** Cloud map of wall thickness distribution of titanium plate and perforated titanium sheet.

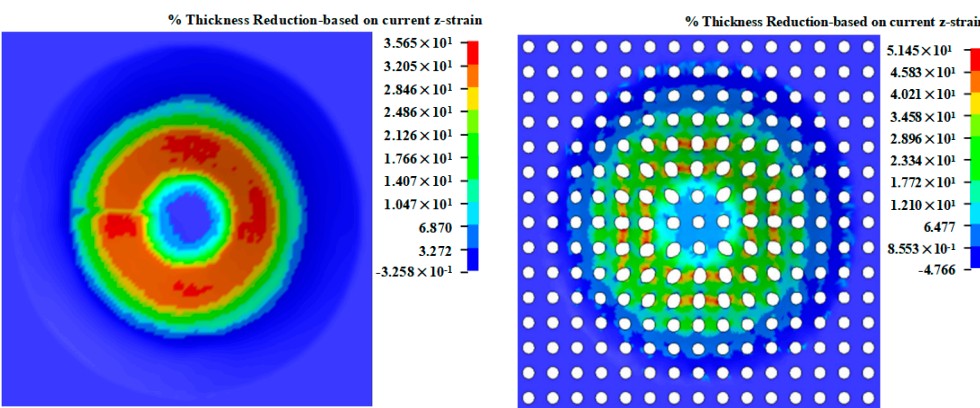

**Figure 11.** Cloud image of the thinning rate of titanium plate and perforated titanium sheet.

Except for the above three special areas, it can be observed from the cloud map that the sinusoidal law is used in the incremental forming process., When the inclination angle of the cone forming is relatively low, the wall surface is uniformly thinned, and the thickness of the perforated titanium sheet changes in line with the sinusoidal law. With the increase of the forming inclination, the inhomogeneous thinning occurs, resulting in local thinning of the discontinuous area of the cone wall of the perforated titanium sheet. In the area of four mesh apertures away from the mesh area, the sheet is consistent with the deformation of the short shaft side.

As presented in Figure 11, the elements of the perforated titanium sheet Y = 0 cross–section (X axis) are discussed in two regions: region I is a continuous element along the wall direction, as presented in Figure 12 with BC, EF, HI, KL, B′C′, E′F′, H′I′, K′L′ sides of the elements. Area *II* is a discontinuous element on the perforated titanium sheet domain side.

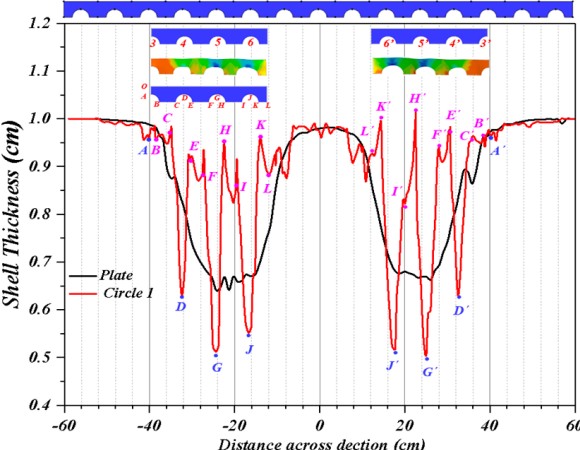

**Figure 12.** Y = 0 wall thickness distribution curve of titanium plate and perforated titanium sheet cross-section.

The strain state of the element of the perforated titanium sheet in the region *I* is radially tensed, the circumferential tension, the thick direction is pressurized, and the $\varepsilon_1$, $\varepsilon_2$, $\varepsilon_3$, $\bar{\varepsilon}$, and energy density of most of the elements is smaller than that of the region *II* element, so the wall thickness is larger than that of the region *II* element. The strain state of the element of the perforated titanium sheet in Area *II* is radial tension, circumferential compression, and thick compression. Perforated titanium sheet domain is structured so that in the discontinuous area of the metal material in free space, the minimum resistance of flow is the smallest. The metal material flows along the mesh in the direction of the wall, therefore, in the discontinuous region of the element, deformation is larger, the strain curve of each circular aperture has a "finger–like" protrusion, and each "finger–like" protrusion corresponds to the short axial side element (D, G, J, D′, G′, J′) at that location $\varepsilon_2 \approx 0$,

$\varepsilon_1 \approx -\varepsilon_3$. The "finger–like" protrusion indicates that the absolute value of energy density and strain at this location is large, so the thinning rate of the perforated titanium sheet is also the largest.

### 3.4. Displacement Field

In the early stages of the incremental forming process, displacement occurs mainly along the Z–axis. However, as the incremental forming process progresses, under the influence of the friction between the tool and the plate, the metal has a certain plastic flow along the circumferential direction, and that part will undergo a certain degree of torsion, as presented in Figures 13 and 14. When the tool pushes the radial outward material towards the cone wall, for the bottom of the truncated cone piece under the tool is doing a rigid translation, the contour radius becomes smaller, and the material at the bottom of the truncated cone piece will also be plastically deformed to the cone wall. In this process, a small outward radial displacement can be observed. The sine law assumes that the displacement of all deformation elements is completely thinned along the Z direction, but the incremental forming process of the perforated titanium sheet produces radial outward displacements except for the upper surface layer, which makes the final thickness of the material slightly smaller than the thickness obtained by the sinusoidal law.

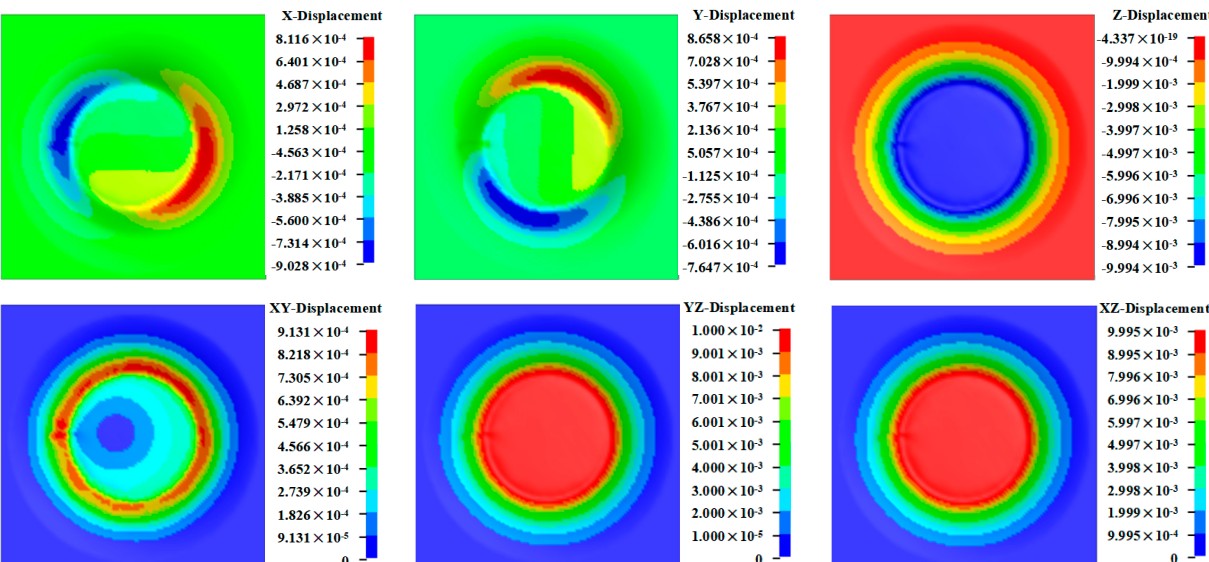

**Figure 13.** Cloud map of titanium plate displacement field.

At the critical x = −40 mm generated at the beginning of the deformation, the X-direction deformation of the titanium plate and the perforated titanium sheet is −0.32 mm and −0.29 mm, respectively, and the Z–direction contour deviation is 3.08 mm and 2.86 mm, respectively, and the perforated structure effectively improves the bending of the unformed free zone at the edge.

The ideal deformation termination should be at the coordinate point (x, z) is (30, −10), but the actual titanium plate and perforated titanium sheet deformation terminates at (30.77, −8.96) and (30.19, −9.39), respectively, the X–direction contour deviation of titanium plate and perforated titanium sheet is 0.77 mm and 0.19 mm, respectively, and the Z–direction contour deviation is 1.04 mm and 0.62 mm, respectively. The lower bottom surface is not formed, the titanium plate produces obvious internal arching phenomenon, and the perforated structure effectively improves the non–forming area of the lower bottom surface of the plate. Regarding the head–in–pillow effect, the z–direction deformation of the two "pillow bags" is −9.27 mm and −9.37 mm, respectively, that is, the Z–direction profile deviation of titanium plate and perforated titanium sheet is 0.73 mm and 0.63 mm, respectively.

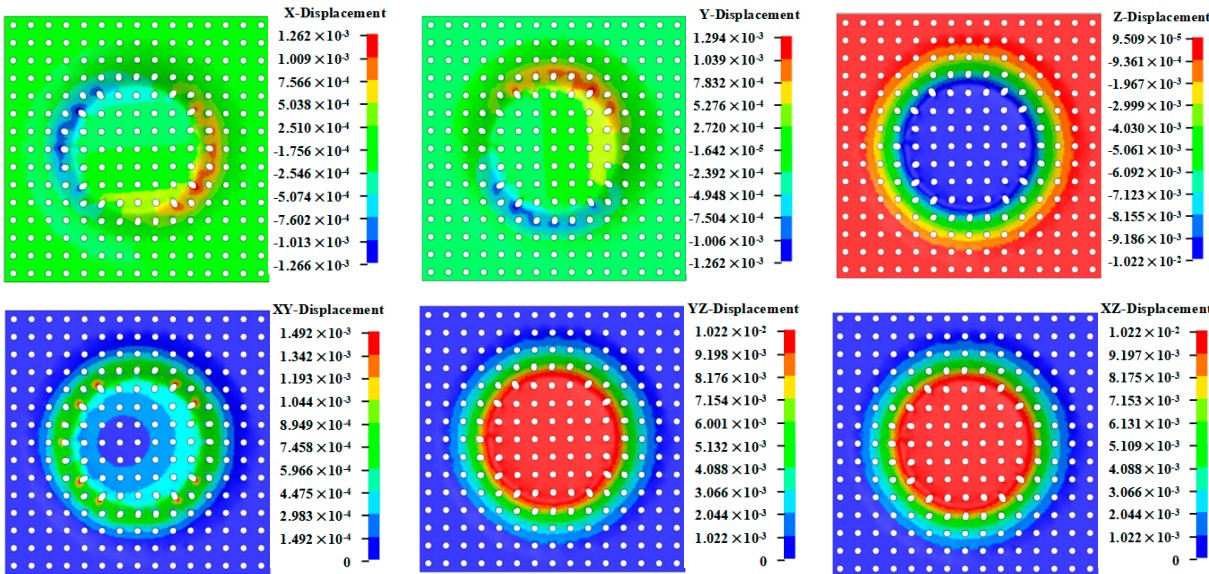

**Figure 14.** Cloud map of perforated titanium sheet displacement field.

The results manifest that the titanium plate produces extra–surface deformation during the incremental forming process, and the aperture structure of the perforated titanium sheet makes the perforated titanium sheet element deformed on the surface where the aperture area is located, and the deformation outside the surface is much smaller than that of the titanium plate, which effectively improves the contour accuracy.

In order to obtain an in–depth understanding of the SPIF deformation behavior and the displacement field evolution of perforated titanium sheet and titanium plate, this paper selects the contour forming path with a Z–direction feed of 1 mm, and uses LS–DYNA to simulate the SPIF process of a truncated cone piece with an opening diameter of 80 mm. Along the horizontal x direction, the simulation information of 5 element s A, B, C, D, and E is extracted from the upper bottom boundary of 3 mm, 5 mm, 6 mm, 10 mm, and 11 mm, respectively (corresponding to the Z–direction ideal displacements of 3 mm, 5 mm, 6 mm, 10 mm, and 11 mm, respectively. Element A is selected in the bending transition deformation area, B, C, D are the three elements in the main deformation area along the radial direction, and element E is located in the transition zone between the inclined wall part and the bottom. The A and D element s of the perforated titanium sheet are located in the lower part of the discontinuous area, the C element is located in the upper part of the discontinuous area, the B and E element s are located in the continuous area, and the E element is located in the transition area at the bottom of the inclined wall and the truncated cone piece.

As presented in Figure 15, the elements located in the unformed area will produce a more obvious pre-deformation before being pressed down by the tool and transferred into the self-forming area. The perforated titanium sheet elements A, B and C are deformed 7, 4 and 2 times, respectively, resulting in their secondary deformation accumulation of 2.03 mm, 1.225 mm, and 0.645 mm, respectively. Element A, B, and C of the titanium plate underwent 7, 4, and 3 times of deformation, respectively, resulting in their secondary deformation accumulation of 1.68 mm, 0.88 mm, and 0.58 mm, respectively, from the ideal value. The formed area has different degrees of pre-deformation on the unformed area. When the tool slides on the inner layer of the titanium plate and perforated titanium sheet, each incremental deformation will only cause the deformation of the contact area and macroscopic embodiment of the Z displacement to increase. When the tool diverges from the area, the deformation no longer increases. The undeformed region of the titanium plate and perforated titanium sheet element with the increased distance from the formed contact areaand the degree of pre-deformation gradually attenuate.

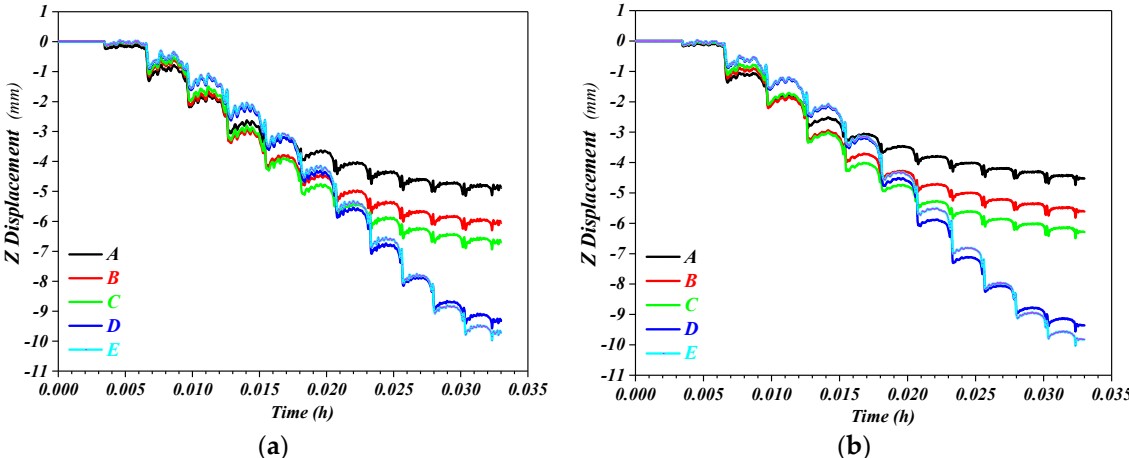

**Figure 15.** Displacement time history diagram of titanium plate and perforated titanium sheet A to E element. (**a**) titanium plate. (**b**) perforated titanium sheet.

In comparison with the formed area of the titanium plate, the element in the same position enters the deformation earlier, and the accumulated secondary deformation is also larger, indicating that the thinning and stretching of the formed area of the perforated titanium sheet is larger than that of the titanium plate.

In respect to the designated elements of titanium plate and perforated titanium sheet, the simulated displacement and ideal displacement, when the tool is detached from the titanium plate and perforated titanium sheet at the end of forming, were extracted respectively, and the displacement deviation of titanium plate and perforated titanium sheet presented in Table 3 is obtained. As can be observed from Table 3, the maximum displacement deviation of the wall to the forming area of the titanium plate during the incremental forming process is 1.88 mm, and the minimum displacement deviation is 0.26 mm. The maximum displacement deviation of the wall forming area of the perforated titanium sheet during the incremental forming process is 1.53 mm, and the minimum displacement deviation is 0.3 mm. It is presented that the aperture structure of the perforated titanium sheet makes the plastic deformation larger than that of the titanium plate. Regarding the displacement of element D at the edge of the circular truncated cone and the element E at the bottom of the truncated cone piece, the ideal displacement deviation of the titanium plate element D and E from the final molding is −0.83 mm and −0.4 mm. Meanwhile, the perforated titanium sheet is −0.67 mm and −0.22 mm, which indicates that the perforated titanium sheet has higher geometric accuracy than the titanium plate.

**Table 3.** Titanium plate and perforated titanium sheet displacement deviation.

| Element | Ideal Displacement (mm) | Simulation Displacement (mm) | | Deviation Displacement (mm) | | Deviation Rate | |
|---|---|---|---|---|---|---|---|
| | | Titanium Plate | Perforated Titanium Sheet | Titanium Plate | Perforated Titanium Sheet | Titanium Plate | Perforated Titanium Sheet |
| A | 3 | 4.88 | 4.53 | 1.88 | 1.53 | 38.52% | 33.77% |
| B | 5 | 6.05 | 5.61 | 1.05 | 0.61 | 17.36% | 10.87% |
| C | 6 | 6.69 | 6.29 | 0.69 | 0.29 | 10.31% | 4.61% |
| D | 10 | 9.32 | 9.36 | −0.68 | −0.64 | −7.3% | −6.84% |
| E | 10 | 9.71 | 9.82 | −0.29 | −0.18 | −2.99% | −1.83% |

## 4. Conclusions

Because of the perforated titanium sheet truncated cone piece model, the finite element all-parameter simulation method was adopted in this paper to explore the formation mechanism of the perforated titanium sheet component in the shape of the truncated cone piece. In the process of incremental forming by means of analyzing the mesh aperture

stress-strain and the distribution law of the aperture diameter growth rate in different directions, the specific conclusions obtained were as follows:

(1) In the bending deflection area, both sides of the short axis of the circular aperture in the main deformation area as well as in the side element of mesh aperture away from the main deformation area, the strain is $\varepsilon_2 \approx 0$, $\varepsilon_1 \approx -\varepsilon_3$, which can be regarded as being subject to shearing action and the state of plane stress. At the bottom of the truncated cone piece and in the side element of the long axis of the circular aperture in the main deformation area, the strain is $\varepsilon_2 > 0$, $\varepsilon_1 + \varepsilon_2 \approx -\varepsilon_3$, which is considered to be in a mechanical state of typical biaxial tension coupled with uniaxial compression.

(2) The strain rate and radial aperture growth rate of perforated titanium sheet are relatively larger than that of the titanium plate. The aperture diameter of perforated titanium sheet gives priority to radial growth, the maximum radial aperture growth rate value is as high as 78.53 percent, the maximum circumferential aperture growth rate value is only 10.84 percent, and the circumferential aperture growth rate is much smaller than the radial growth rate.

(3) The titanium plate undergoes out-of-plane deformation during the incremental forming process, and the aperture structure of the perforated titanium sheet allows the element to deform on the surface where the aperture area is located, and the out-of-plane deformation generated is much smaller than that of the titanium plate, which effectively enhances the contour accuracy. The element in the same position undergoes the deformation earlier in comparison with the formed area of titanium plate, and the accumulated secondary deformation is also larger, indicating that the extension-thinning for the formed area of perforated titanium sheet is larger than that of titanium plate.

**Author Contributions:** Conceptualization, R.L.; methodology, R.L. and T.W.; software, R.L.; experiment and validation, R.L. and T.W.; formal analysis, R.L.; investigation, R.L. and T.W.; data curation, R.L.; writing—original draft preparation, R.L.; writing—review and editing, R.L. and T.W. All authors have read and agreed to the published version of the manuscript.

**Funding:** This research was financially supported by the Key Program for International S&T Cooperation Projects of China (2018YFE0194100) and the Priority Academic Program Development of Jiangsu Higher Education Institutions (2018YFE0194100).

**Data Availability Statement:** Not applicable.

**Conflicts of Interest:** The authors declare no conflict of interest.

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
