# Peer review of "Research on Single Point Incremental Forming Characteristics of Perforated TA1 Sheet"

_metals, doi:10.3390/met12111944_

Round 1

Reviewer 1 Report

This paper deals with single point incremental forming of a titanium alloy sheet with many round holes. It utilizes finite element simulations to investigate the forming characteristics of the sheet with holes comparing with the one without holes.

1. It is not adequate to call the sheet like Figure 4 as "mesh". Usually, ratio of the area of holes is much higher in case of mesh, and structure of a mesh is composed of wires or thin components like wires. The sheet of Fig. 4 is commonly called "perforated metal sheet". Hence the title of the paper must be changed to "Research on Single Point Incremental Forming Characteristics of Perforated TA1 Sheet".

2. Survey on the previous works in the Introduction is quite insufficient. There have been many studies on forming of perforated metal sheet. Those references such as the following list should be properly cited. In particular, Ref. [A] is deeply related with the study of this paper and should be introduced in detail.

[A] Bouzidi, S., Ayadi, M., & Boulila, A. (2022). Feasibility Study of the SPIF Process Applied to Perforated Sheet Metals. Arabian Journal for Science and Engineering, 1-28. https://www.researchgate.net/profile/Atef-Boulila/publication/358646969_Feasibility_Study_of_the_SPIF_Process_Applied_to_Perforated_Sheet_Metals/links/620d570a6c472329dcebdbbb/Feasibility-Study-of-the-SPIF-Process-Applied-to-Perforated-Sheet-Metals.pdf

[B] Elangovan, K., & Narayanan, C. (2010). Application of Taguchi approach on investigation of formability for perforated Al 8011 sheets. International Journal of Engineering, Science and Technology, 2(5), 300-309. https://www.ajol.info/index.php/ijest/article/view/60170/48418

[C] Baik, S. C., Han, H. N., Lee, S. H., Oh, K. H., & Lee, D. N. (1997). Plastic behaviour of perforated sheets under biaxial stress state. International journal of mechanical sciences, 39(7), 781-793. https://doi.org/10.1016/S0020-7403(96)00091-4

3. The method of the FEM simulation should be described in more detail. For example, how was the material of the sheet divided into the elements? What shape and size were those elements and how were they arranged?

4. Perforating metal sheet involves plastic deformation around the punched holes. Was the shape of holes in the FEM model same as the real perforated sheet? Punching process of the perforated metal sheet might induce work hardening and residual stress around the holes. Was the influence of such changes from the blank sheet considered in this study?

5. Contribution of the paper seems rather weak because it just shows predictions by the FEM simulation and does not compare the simulation results with the real experimental results for verification. Moreover, the paper uses only one combination of the forming parameters and does not cover the various processing conditions that can be selected. It would be preferable at least either experimental verification or forming parameter analysis is incorporated.

Author Response

Dear editors and reviewers,

     Thank you for your letter and for the reviewers’ comments concerning our manuscript entitled “Research on Single Point Incremental Forming Characteristics of TA1 Mesh ” (Metals-1991441).
Those comments are all valuable and very helpful for revising and improving our paper, as well as the important guiding significance to our researches. We have modified and added the manuscript accordingly. Detailed corrections have been listed below point by point. In addition, we highlighted the changes with red colored text.

Reviewer 2 Report

I make some revision suggestions below, but the authors need to make a significant effort to better interpret and explain their results.

1-     The paper will need language and style editing for further improvement.

2-     What is your novelty? Please, explain your novelty.

3-     Please used the quantity results at the end of your abstract.

4-     Some of the most relevant papers in the single point incremental forming are not cited, for instance:

https://doi.org/10.3390/met11081188
https://doi.org/10.1016/j.compositesa.2020.106209
https://doi.org/10.1016/j.jmatprotec.2021.117365
https://doi.org/10.3390/ma14216372
https://doi.org/10.1016/j.jmrt.2020.12.014
https://doi.org/10.1016/j.jmapro.2020.12.022
https://doi.org/10.1007/s12289-017-1387-y
https://doi.org/10.1016/j.ijmecsci.2017.12.053
https://doi.org/10.1016/S1003-6326(12)61683-5
https://doi.org/10.1016/j.ijlmm.2019.04.003
https://doi.org/10.1007/s00170-018-3148-6

Author Response

(The authors gave the same response as above.)

Reviewer 3 Report

Dear Authors,

Congratulations on your work, which is focused on a very interesting subject. As any other paper in this phase, there are some amendments to do, whose can improve the overall quality of your paper. Thus, I'm providing below some comments and suggestions, trying to collaborate by this way in improving your paper:

1. The Abstract doesn't clearly state the literature gap found, as well as the main motivation to develop this work. Thus, please clearly state the gap found in the literature in the Abstract, Introduction and Conclusions. The main goals are also unclear in the Abstract.

2. The novelty brought by your work is also not properly pointed out. Thus, please state clearly the novelty that your paper represents for the scientific community, stating as well if your contribution is exclusively scientific or if there was some practical motivation behind the development of your work. Any industrial application based on this work should also be pointed out.

3. The references in the Literature Review don't follow the format required by the journal. Thus, please adapt them to the journal template.

4. Literature Review is well done, but readers prefer direct speech, describing briefly in what the work of previous Researchers has been focusedon, methodology used and main results. Please avoid as much as possible generic ideas.

4. Please avoid a large group of references under the same idea [6-13].

5. Please use always a free space between units and values (0.5 mm, 2 mm/min for example).

6. Please point out the standards used to characterize the samples (Tensile tests, and so on).

7. Please correct the values of the Young's Modulus for the Titanium, because they are not expressed in MPa, but in GPa.

8. Please provide details about the tool used, namely, material, geometry, friction coefficient, and so on.

9. Please provide details about the boundary conditions considered.

10. Please provide information about the gripping system considered.

11. Please provide information about the number of elements/nodes used.

12. The detail in Figure 5 is not enough to analyse what you intend to show. Please improve the images quality and size.

13. Please detail the caption of Figure 7, as well as confirm the units used (maybe there is a mismatch with the others considered in the text body).

14. No validation of the simulation is shown, as evidence. Please complete.

15. No discussion of the results is presented. Please include a Discussion section, comparing your results with others previously obtained.

Best wishes.

Kind regards.

Author Response

(The authors gave the same response as above.)

Round 2

Reviewer 1 Report

The authors have intensively revised their manuscript well reflecting each comment of this reviewer in the first round. Please correct the following points.

1) The newly added photographs of page 10, which might be the SPIF experimental results of titanium plate and perforated sheet, should be explained in the caption of Figure 9 and it should be mentioned in the text.

2) The title of Ref. [22] is incorrect (Page 18, line 568). It is "Feasibility Study of the SPIF Process Applied to Perforated Sheet Metals". The family name of the first author is "Bouzidi" (Page 2, line 96).

2) Page 2, line 86: "the the" -> "the"

3) Page 10, line 312: what is "the weekly aperture size growth rate"?

4) Page 1, line 14: "qualifification" -> "qualification" (Please use spell checker)

Author Response

Dear editors and reviewers,

Thank you for your letter and for the reviewers’ comments concerning our manuscript entitled “Research on Single Point Incremental Forming Characteristics of TA1 Mesh ” (Metals-1991441).

Those comments are all valuable and very helpful for revising and improving our paper, as well as the important guiding significance to our researches. We have studied comments carefully and have made correction which we hope meet with approval. Revised portion are marked in red in the paper. The revision was addressed point by point below. In addition, we would like to make a modification to our list of authors to reflect their separate contributions to the new version of the paper.

Reviewer 3 Report

Congratulations for the improvements performed.

Kind regards.

Author Response

(The authors gave the same response as above.)
